# Near-field transmission matrix microscopy for mapping high-order eigenmodes of subwavelength nanostructures

Eunsung Seo[1,2,5], Young-Ho Jin [3,5], Wonjun Choi [1,2], Yonghyeon Jo [1,2], Suyeon Lee[4], Kyung-Deok Song[1,2], Joonmo Ahn[1,2], Q.-Han Park[2], Myung-Ki Kim [3✉] & Wonshik Choi [1,2✉]

As nanoscale photonic devices are densely integrated, multiple near-field optical eigenmodes take part in their functionalization. Inevitably, these eigenmodes are highly multiplexed in their spectra and superposed in their spatial distributions, making it extremely difficult for conventional near-field scanning optical microscopy (NSOM) to address individual eigenmodes. Here, we develop a near-field transmission matrix microscopy for mapping the high-order eigenmodes of nanostructures, which are invisible with conventional NSOM. At an excitation wavelength where multiple modes are superposed, we measure the near-field amplitude and phase maps for various far-field illumination angles, from which we construct a fully phase-referenced far- to near-field transmission matrix. By performing the singular value decomposition, we extract orthogonal near-field eigenmodes such as anti-symmetric mode and quadruple mode of multiple nano-slits whose gap size (50 nm) is smaller than the probe aperture (150 nm). Analytic model and numerical mode analysis validated the experimentally observed modes.

[1] Center for Molecular Spectroscopy and Dynamics, Institute for Basic Science, Seoul 02841, Korea. [2] Department of Physics, Korea University, Seoul 02841, Korea. [3] KU-KIST Graduate School of Converging Science and Technology, Korea University, Seoul 02841, Korea. [4] Samsung Advanced Institute of Technology, 130, Samsung-Ro, Yeongtong-Gu, Suwon, Gyeongi-Do 16678, Korea. [5]These authors contributed equally: Eunsung Seo, Young-Ho Jin. ✉email: rokmk@korea.ac.kr; wonshik@korea.ac.kr

**V**arious types of resonant optical interactions with nanostructures have been exploited in nanophotonics to engineer the performance of nanoscale devices for applications ranging from sensing[1] to optoelectronics and metamaterials[2–4]. By tailoring the collective optical modes associated with the resonant interactions[5], important figure of merits such as the coupling efficiency of the far-field illumination to the near-field, local field enhancement, spatial field localization, and control of nanoscale light flow[6] have been significantly improved. In these studies, NSOM has served as a general and robust tool to map the local near-field for unveiling the underlying process of nanoscopic physical phenomena and optimizing the performance of nanophotonic devices[7]. It has been especially useful for visualizing the formation of these optical modes and their coupling with the other spatially and spectrally overlapped modes[8–12].

In NSOM, subwavelength probes are used to convert the near-field evanescent waves carrying spatial frequencies exceeding the far-field limit set by $k_0 = 2\pi/\lambda$, where $\lambda$ is the wavelength of the light source[13], to the detectable far-field waves or vice versa. To map the optical eigenmodes in nanostructures, the probe is placed near the device to either collect local near-field electromagnetic wave or launch a spatially localized electromagnetic wave[8]. Importantly, the excitation wavelength is tuned to the resonance of the particular optical mode for selectively mapping the target modes. This approach has been widely used to analyze the plasmonic and photonic modes of various systems[7,14,15].

With advancements in fabrication technology, there have been growing interest in designing smaller nanostructures and integrating them at a greater density for merging multiple functionalities within a smaller chip[7,11]. This often makes it necessary to engineer the hybridized modes having distinct resonances with minimal cross-talks by properly engineering the coupling among the modes of the constituent nanostructures[12,16,17]. However, mode hybridization is more complicated because the modes of the basic building blocks such as nanoparticles, nanorods and nano-slits are spectrally broadened and spatially overlapped with the reduction of their size[18]. In this respect, most of the existing NSOM modalities are not well suited for elucidating the detailed mode formation mechanism. They tend to map either the dominant modes or integration of all the modes at the given excitation/detection wavelength. To better understand the formation of the hybridized modes in the small-scale nanophotonic devices, it will be important to have experimental tools that can separately map the multiple superposed modes.

Here, we proposed a near-field imaging method termed a near-field transmission matrix microscopy for simultaneously mapping multiple hybridized eigenmodes that are formed by the coupling between the modes of the constituent nanostructures. We constructed an interferometric NSOM system based on the self-interference geometry and measured both the amplitude and phase maps of the near-field wave at the surface of nanostructures. Various phase imaging NSOM techniques have been developed in the past[14,19–21], but here we made an important addition. We scanned the angle of the far-field illumination and recorded the near-field complex-field maps for various illumination angles. These measurements allow us to construct a fully phase-referenced far- to near-field transmission matrix (FNTM), which describes the far-field input to near-field output response of the given nanostructures. By performing the singular value decomposition (SVD) of the measured matrix, we could extract individual orthogonal near-field eigenmodes contributing to the input–output response of the device. With this approach, multiple high-order modes of the double nano-slits whose spacing (50 nm) is significantly smaller than the diameter of the NSOM probe aperture (150 nm) were made visible. This is an important addition of information to the conventional NSOM, in which

only the integrated signal from all the contributing modes at the excitation wavelength is detected. We provided a theoretical model to support the working principle of the proposed method and conducted numerical mode analysis to validate the experimentally observed modes. In addition, we found that the mapping of the antisymmetric mode allows us to better resolve the existence of multiple nano-slits whose gap size is smaller than the probe aperture.

## Results

**Experimental measurement of a FNTM.** A transmission matrix describes the coherent linear interaction between light and arbitrary device including disordered media[22,23]. It describes the complex-field map, i.e., phase and amplitude maps, at the output plane of the device depending on the excitation of free modes at the input plane. It has been widely used in the far-field regime in the past and offered unique opportunities. For example, the inversion of the transmission matrix led to image delivery through a scattering layer[24], and the eigenmodes of the transmission matrix were used for efficient light energy delivery and mapping target objects within the scattering medium[24–28]. The transmission matrix approach also allows for the control and image delivery of near-field waves[29–32]. In the present study, we developed an experimental method to record a fully phase-referenced FNTM, $t(x, y; \mathbf{k}_{\text{in}})$, which describes the near-field complex-field maps at the upper surface $(x, y, z=0)$ of the nanostructures for the far-field illumination at the bottom surface $(z = -z_0)$ with various transverse wavevectors, $\mathbf{k}_{\text{in}} = (k_{\text{in}}^x, k_{\text{in}}^y)$, set by the illumination angle. Figure 1 shows a simplified experimental scheme where an NSOM aperture probe picked up the near-field wave at the upper surface (see also Supplementary Note 1 for the detailed experimental setup). The amplitude and phase of the near-field wave were recorded by using the self-interference phase-shifting interferometry. To achieve this, we installed a spatial light modulator (SLM) at the conjugate plane to the sample plane in the illumination beam path and wrote a phase pattern on the SLM to generate two plane waves at the bottom of the nano-slits, one with a normal incidence angle (i.e., $\mathbf{k}_{\text{in}} = 0$) and the other with nonzero $\mathbf{k}_{\text{in}}$

$$E_{\text{in}}\left(x, y, z = -z_0; \mathbf{k}_{\text{in}}, \Delta\phi_{\text{R}}\right) = A_0 e^{-i\Delta\phi_{\text{R}}} + A_0 e^{-i\left(k_{\text{in}}^x x + k_{\text{in}}^y y\right)}. \quad (1)$$

Here, the normally incident plane wave, whose wavefronts are indicated by the red lines in Fig. 1a, serves as a reference wave. Its relative phase $\Delta\phi_{\text{R}}$ with respect to the other sample wave, indicated by the blue lines in Fig. 1a, was controlled by the phase pattern on the SLM. The transmitted wave on the upper surface of the nanostructures can be expressed as

$$E_{\text{out}}\left(x, y, 0; \mathbf{k}_{\text{in}}, \Delta\phi_{\text{R}}\right) = E_{\text{R}}(x, y)e^{-i\Delta\phi_{\text{R}}} + E_{\text{S}}(x, y; \mathbf{k}_{\text{in}}), \quad (2)$$

where $E_{\text{R}}(x, y)$ and $E_{\text{S}}(x, y; \mathbf{k}_{\text{in}})$ are the complex-field amplitudes of the reference and sample waves, respectively, at the upper surface. The NSOM probe recorded the interference intensity $I_{\text{out}} = \left|E_{\text{out}}\left(x, y, 0; \mathbf{k}_{\text{in}}, \Delta\phi_{\text{R}}\right)\right|^2$, which is expressed as a sinusoidal function of $\Delta\phi_{\text{R}}$ (see the example in Supplementary Note 1). By measuring the intensities at four incremental steps of $\Delta\phi_{\text{R}}$, i.e., $\Delta\phi_{\text{R}} = 0, \frac{\pi}{2}, \pi$ and $\frac{3\pi}{2}$, we could demodulate the amplitude and phase of $E_{\text{S}}(x, y; \mathbf{k}_{\text{in}})$[33]. Here, we accounted for the amplitude of $E_{\text{R}}(x, y)$ by separately measuring its intensity map taken only for the normal illumination. The phase of $E_{\text{R}}(x, y)$ was assumed to be flat in space, which is typical for the symmetrically driven subwavelength nanostructures (see Supplementary Note 3). In the experiment, we chose an illumination wavevector $\mathbf{k}_{\text{in}}$ and sequentially displayed four phase patterns on the SLM setting $\Delta\phi_{\text{R}}$ to multiples of $\frac{\pi}{2}$. The NSOM probe was scanned across the lateral plane at the upper surface as shown in the left image of

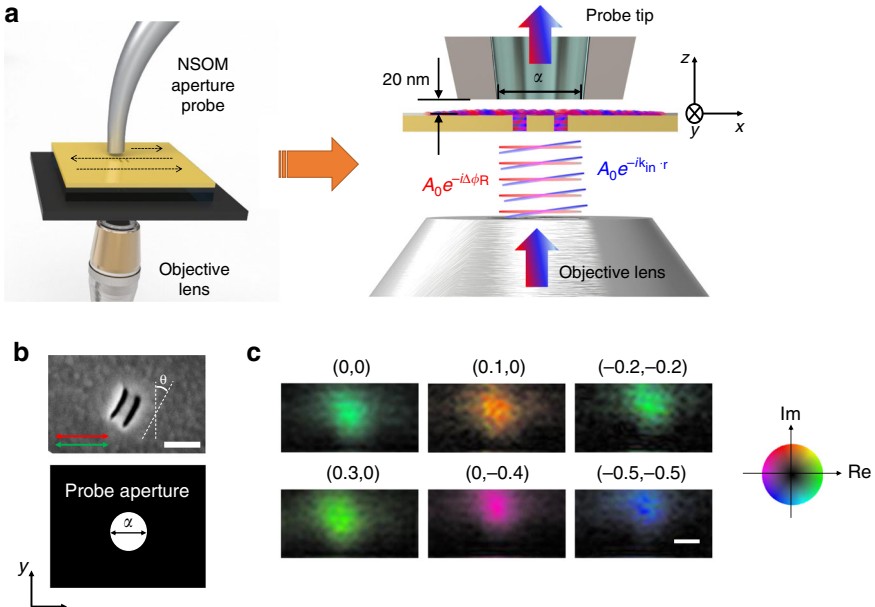

**Fig. 1 Experimental recording of a FNTM. a** Simplified experimental schematic diagram. Light waves were introduced from the bottom of the sample through the objective lens, and the aperture-type probe was scanned to obtain near-field maps. For the recording of a FNTM, two planar waves, $A_0 e^{-i\Delta\phi_R}$ (red lines) and $A_0 e^{-i\mathbf{k}_{in}\cdot\mathbf{r}}$ (blue lines), generated by the SLM (not shown) were sent through the objective lens to the bottom of the nano-slits (image on the right). An aperture probe made of a tapered fiber coated with a gold layer was positioned close to the upper surface. The diameter of the aperture $\alpha$ was 150 nm. The probe converted the interfered near-field wave to the far-field wave, which was then delivered to the photodetector (not shown). **b** Upper image: scanning electron micrograph of the double nano-slits. Scale bar, 150 nm. The red arrow indicates the polarization of the far-field illumination, and the green arrow shows the polarization of the near-field collection set according to the fiber probe bending direction. The nano-slits were rotated by $\theta =$ 34° clockwise. Lower image: the dimension of the probe aperture used in the experiment. **c** Complex-field maps of the near-field waves acquired by NSOM for various incident wavevectors, $\mathbf{k}_{in}$. The coordinate above each sub-figure indicates $\mathbf{k}_{in} = \left(k_{in}^x, k_{in}^y\right)$ in unit of $k_0$. Scale bar, 150 nm. Circular color map: real and imaginary values of the near-field wave.

Fig. 1a to obtain four corresponding near-field intensity maps. By applying the phase-shifting interferometry algorithm, we obtained the near-field complex-field map for the corresponding $k_{in}$, i.e., $E_S(x, y; \mathbf{k}_{in})$, and used it to construct the FNTM.

To validate the proposed method, we prepared a pair of nano-slits whose characteristic physical dimensions are significantly smaller than the diameter of the NSOM probe aperture (100 or 150 nm). Nano-slits are one of the fundamental building blocks of complex devices[12,34]. The formation of hybridized eigenmodes by the mode coupling between the nearby nano-slits was well understood both theoretically and experimentally[34,35]. Therefore, devices composed of nano-slits can serve as ideal platforms for validating the proposed method. We fabricated double nano-slits on a 100-nm-thick gold film by the focused ion beam milling, as shown in the upper image in Fig. 1b (see Supplementary Note 4 for the sample fabrication details). The width $W$ and length $L$ of each slit were approximately 20 and 160 nm, respectively. The gap $D$ between the two slits was approximately 50 nm. The long axis of the nano-slits was rotated by $\theta = 34°$ clockwise with respect to the vertical line, and the polarization direction of the far-field illumination was set to $x$-direction (red arrow in Fig. 1b). In this arrangement, both the transverse and longitudinal modes with respect to the long axis of the nano-slits were excited. The aperture diameter $\alpha$ of the NSOM probe was 150 nm, which was much smaller than $\lambda_{exc} = 637$ nm, but three times larger than the slit gap. The NSOM probe was made of an Au–Cr coated tapered waveguide, and its aperture dimension is shown in Fig. 1b for the direct comparison with the dimension of the nano-slits. The bending direction of the tapered fiber probe used in the experiment determines the polarization of near-field collection, which is indicated by the green arrow in Fig. 1b[36]. Similar to the

excitation geometry, this allows the detection of both the transverse and longitudinal modes. For each $\mathbf{k}_{in}$, we scanned the NSOM fiber probe around the center of the nano-slits over an area of $800 \times 400$ nm$^2$ with a scanning step of 25 nm. The distance of the NSOM probe to the sample surface was maintained at approximately 20 nm throughout the set of measurements. We repeated the same measurements for 100 different $\mathbf{k}_{in}$, which uniformly covered the full numerical aperture (NA = 0.6) of the bottom objective lens (see Supplementary Note 1 for the detailed coverage of $\mathbf{k}_{in}$). The measurements of the entire angular set of complex-field maps lasted around 27 min. Figure 1c shows the representative near-field complex-field maps for each $\mathbf{k}_{in}$ in unit of $k_0$. These individual NSOM images, each of which corresponds to a conventional NSOM image, mainly visualized the fundamental symmetric mode. Multiple higher order transverse and longitudinal modes were spectrally over-lapped to such a degree that individual modes cannot be excited by the selected excitation wavelength[37]. Therefore, relatively weak higher order modes were obscured by the fundamental mode.

**Extraction of high-order near-field eigenmodes**. The near-field complex-field map (Fig. 1c), $E_S(x, y; \mathbf{k}_{in})$, resulted from the linear superposition of the orthogonal near-field eigenmodes. With the increase of $|\mathbf{k}_{in}|$, the phase difference of the excitation wave at the two nano-slits is increased such that the way the eigenmodes are superposed varies with $\mathbf{k}_{in}$. In this section, the extraction of individual eigenmodes contributing to $E_S(x, y; \mathbf{k}_{in})$ is described. First, we constructed a FNTM, $t(x, y; \mathbf{k}_{in})$, by assigning 100 complex-field maps taken by different values of $\mathbf{k}_{in}$ as constituent columns. It is worth noting that the maintenance of the phase

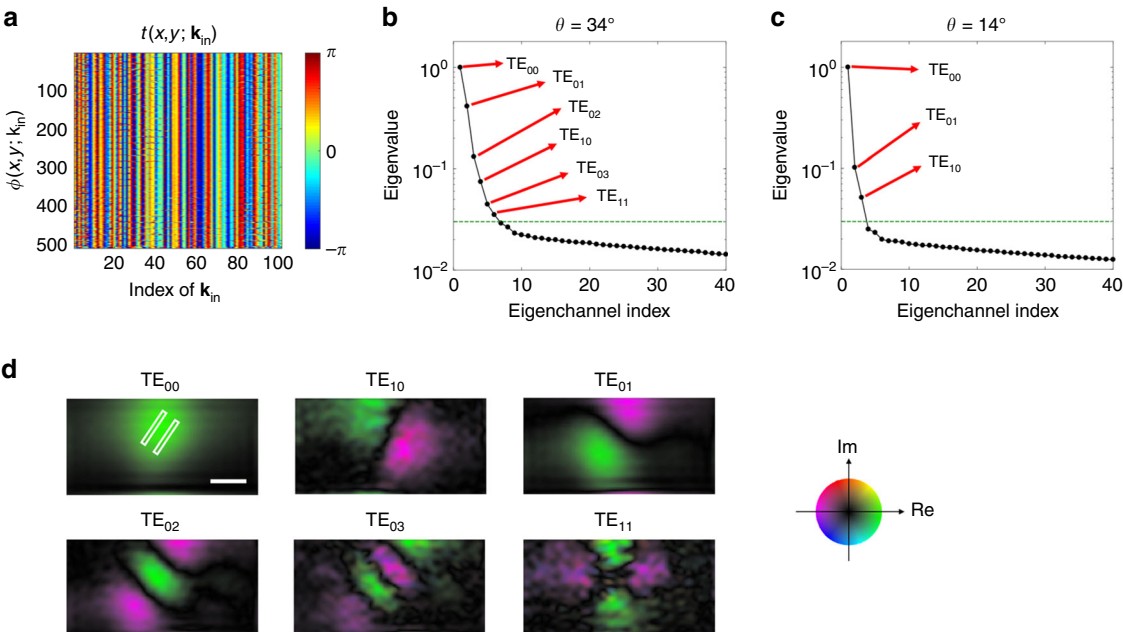

**Fig. 2 Extraction of near-field eigenmodes from the FNTM. a** FNTM constructed from the near-field complex-field maps in Fig. 1c. The column index indicates $\mathbf{k}_{in}$ sorted in increasing order of its magnitude. The row index describes $(x, y)$ sorted in increasing order of $x$ and $y$. Only the phase part of the FNTM is shown. **b** Eigenvalues of $t^{\dagger}t$ in a sorted in the descending order after normalizing them by the largest eigenvalue. The first six eigenvalues were related to the meaningful eigenmodes of the double nano-slits whose indices are indicated by red arrows. **c** Eigenvalues of $t^{\dagger}t$ taken for the sample with $\theta = 14°$. Green dashed lines indicate the noise level of the system. **d** Complex-field maps of output eigenchannels obtained from the first six columns of the unitary matrix $U$ of the FNTM in (**a**). The white rectangles outline the boundaries of the nano-slits. Scale bar, 150 nm. Circular color map: real and imaginary values of the complex field. Output eigenchannels for $\theta = 14°$ are shown in Supplementary Note 1.

stability is crucial as the measurement was performed on a point-by-point basis. As all of the phase measurements were carried out with respect to the normal illumination whose relative phase was well controlled by the SLM, our measurements were sufficiently robust to link multiple measurements together in their phases. This ensured that the measured FNTM was fully phase-referenced. Figure 2a shows the phase part of the FNTM. To identify near-field modes, we performed the SVD of the matrix, i.e., $t(x, y; \mathbf{k}_{in}) = U\Sigma V^{\dagger}$, where $V$ and $U$ are the unitary matrices whose columns are the input and output eigenchannels, respectively, and $\Sigma$ is a diagonal matrix whose diagonal elements are nonnegative real numbers referred to as singular values[23]. $V^{\dagger}$ indicates a conjugate transpose of $V$. The squares of the singular values are the eigenvalues of $t^{\dagger}t$, which are shown in Fig. 2b after sorting them in the descending order with respect to the eigenchannel index. Essentially, SVD could identify the orthogonal basis at the input ($V$) that is mapped onto the orthogonal output basis ($U$) for the linear transformation operator, $t$. Therefore, the columns of $U$ contain near-field eigenmodes at the upper plane of the nano-slits (see Supplementary Note 2 for further explanation). In addition, the physical meaning of the eigenvalue is the coupling efficiency of each mode from the far-field input to the near-field output. We found that the first six largest eigenvalues were above the noise level for the sample with $\theta = 34°$; their eigenchannels were the near-field eigenmodes of the double nano-slits.

In Fig. 2d, we visualized six near-field eigenmodes obtained from the first six columns of $U$. The first column of $U$, associated with the largest eigenvalue, corresponds to the symmetric mode (TE$_{00}$ mode). The mode indices $n$ and $m$ in TE$_{nm}$ indicate the orders of transverse and longitudinal modes, respectively, with respect to the long axis of the nano-slits, and the associated eigenvalue is represented as $\tau_{nm}$. The fourth column of $U$ corresponds to the antisymmetric transverse electric mode (TE$_{10}$

mode), according to the spatial phase distribution and sharp dark line in the middle. Notably, the eigenvalue of TE$_{10}$ mode was 13.4 times smaller than that of the symmetric mode (TE$_{00}$ mode) as shown in Fig. 2b. This explains why it was not visible in the individual near-field maps in Fig. 1c. Based on the shape of this antisymmetric mode, we could estimate the position and the rotation angle of the double nano-slits. The long axis of the slit was rotated by 34° with respect to the polarization of the incident wave, which agrees well with our experimental preparation. The two white rectangular boxes in the TE$_{00}$ mode map indicate the positions and orientations of the slits identified by the antisymmetric mode. Furthermore, we identified other longitudinal high-order modes, the TE$_{01}$, TE$_{02}$, and TE$_{03}$ modes from the second, third, and fifth largest eigenvalues, respectively. These eigenmodes were excited by the polarization component of the illumination along the long axis of the nano-slits. We could even identify the quadrupole mode (TE$_{11}$ mode), whose near-field map shows that the directions of the dipole moments at the two slits were opposite. This mode was rarely observed in conventional near-field imaging as the coupling efficiency of the far-field energy to this mode was extremely low, which was reflected by the extremely small eigenvalue (28.4 times smaller than the eigenvalue of TE$_{00}$ mode). All these high-order modes revealed the local subaperture-scale near-field waves as well as the structural details of the nanostructures that are invisible in the symmetric mode.

We performed a separate measurement for the sample rotated by an angle $\theta = 14°$, smaller than the previous one. The eigenvalue distribution for this sample geometry is shown in Fig. 2c and its eigenmodes are shown in Supplementary Note 1. We observed that the eigenmodes were rotated according to the rotation of the sample. There were two noteworthy observations that support the validity of our measurements. The eigenvalue ratio $\tau_{01}/\tau_{00}$ was reduced from 0.4 to 0.1 as $\theta$ was reduced from

34° to 14°. This is because the excitation and collection of the longitudinal $TE_{01}$ mode were less efficient with the reduction of $\theta$. On the contrary, the eigenvalue ratio $\tau_{10}/\tau_{00}$ was reduced from 0.07 to 0.05, which remained similar with the rotation of the sample as the $TE_{00}$ and $TE_{10}$ modes are both transverse mode. It is noteworthy that the scanning step of the NSOM probe (25 nm) was much finer than the resolving power set by the probe aperture diameter (150 nm)[38] to ensure high mode reconstruction fidelity and minimize the pixelation artifact. From the separate analysis, we confirmed that the scanning step of 50 nm was good enough to map all the observed modes.

The underlying principle of extracting high-order eigenmodes by the SVD of the measured FNTM can be understood by a simple double-slit model, where we considered only the symmetric and antisymmetric modes (see "Methods" and Supplementary Note 2 for the detailed theoretical model). As the slit separation is too small for far-field illumination, i.e., the slit gap $D$ is much smaller than the wavelength, far-field illumination is mainly coupled to the symmetric mode, and the antisymmetric mode is barely excited. For a given far-field incident wavevector $\mathbf{k}_{in}$, the phase difference of the incident wave between the two slits is given by $\Delta\varphi(\mathbf{k}_{in}) = |\mathbf{k}_{in} \cdot \mathbf{D}|$, where $\mathbf{D}$ is a vector connecting the centers of the two slits. As $|\mathbf{k}_{in}| \leq k_0$, $\Delta\varphi \leq \frac{2\pi D}{\lambda} \sim 0.5$, much smaller than $\pi$. Therefore, the incident wave is coupled mostly to the symmetric mode even at the maximum incidence angle. This explains the visibility of only the symmetric mode in conventional NSOM imaging (Fig. 1c) and the eigenvalues of the higher-order modes were tens of times smaller than the symmetric mode (Fig. 2b). In the experiment, the linear combination of these orthogonal modes was measured, and SVD served as the means to identify individual orthogonal modes from the superposed measurements. In the simple double-slit model, we constructed a FNTM after accounting for the far-field excitation limit and analytically demonstrated that SVD can identify the symmetric and antisymmetric modes separately. The eigenvalue ratio $\tau_{10}/\tau_{00}$ between the antisymmetric and symmetric modes is estimated as $\frac{1}{16}\Delta\varphi^2$ in the weak-coupling regime. This model explains the experimentally measured ratio, which was on the order of $10^{-1}$–$10^{-2}$. A more general model incorporating the spatial shapes of the near-field modes was also developed (see Supplementary Note 2) to confirm the capability of our FNTM approach for extracting high-order near-field eigenmodes.

As the eigenvalues of higher-order modes were extremely small in the far-field excitation, it was crucial to increase the sensitivity of the measurements. This was especially the case given the noisy nature of the near-field recording due to weak signal strength. A large number of $\mathbf{k}_{in}$ measurements were important in this instance. Although it is sufficient for the number of $\mathbf{k}_{in}$'s to be equal to the number of orthogonal modes in noise-free measurements, we used 100 different $\mathbf{k}_{in}$ values for the matrix measurement, far larger than the required number of measurements. This increase in the number of independent measurements increased the fidelity of mode mapping, thereby enhancing the signal-to-noise ratio, particularly for the mapping of higher-order modes. To confirm this, we constructed multiple FNTMs by varying the number $N_{in}$ of $\mathbf{k}_{in}$ measurements used for matrix construction. By performing the SVD of each matrix with $N_{in}$ columns, we identified the observable near-field eigenmodes. As shown in Fig. 3, the FNTM with $N_{in} = 2$ showed only the $TE_{00}$ mode because the eigenvalue of $TE_{01}$ was smaller than the experimental noise level. In contrast, the FNTM with $N_{in} = 10$ columns revealed the $TE_{01}$ mode. When we increased $N_{in}$, various higher-order near-field modes were observed as the eigenvalues of the corresponding eigenmodes were increased beyond the noise

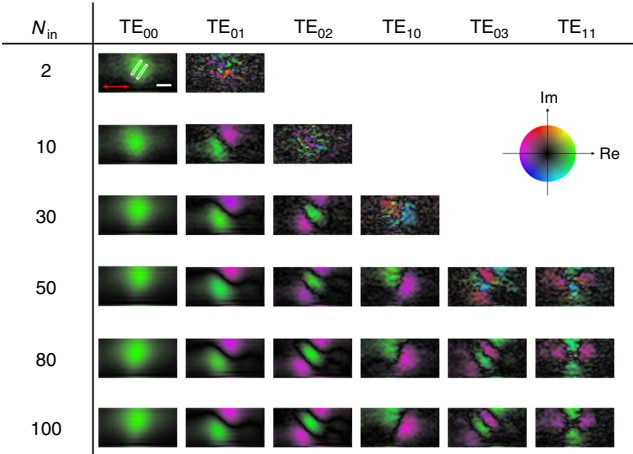

**Fig. 3 Sensitivity of near-field mode mapping depending on the number of incidence angles.** Among the 100 different $\mathbf{k}_{in}$ measurements, $N_{in} = 2$, 10, 30, 50, 80, and 100 measurements were used to construct FNTMs, and the corresponding near-field eigenmodes are shown. Circular color map: real and imaginary values of the complex field. The white rectangles outline the boundaries of the nano-slits. Scale bar, 150 nm.

level. Observing the $TE_{02}$ mode extracted in the $N_{in}$ range of 10–100, we could recognize that the image quality was improved with the increase in $N_{in}$. This clearly supports that a large number of $\mathbf{k}_{in}$ measurements played a crucial role in identifying the near-field eigenmodes. Conventional NSOM corresponds to $N_{in} = 1$ case for the plane wave excitation or coherent integral of all the measurements for the focused illumination, which explains why it is unable to extract individual modes.

**Numerical validation of the experimentally observed modes.** We validated the experimentally observed near-field eigenmodes by numerical mode analysis based on the finite-difference time-domain (FDTD) method. As shown in Fig. 4a, the geometric parameters of the sample were taken from the fabricated sample ($L = 160$ nm, $W = 20$ nm, and $D = 50$ nm). Here, the double nano-slits were prepared on a 100-nm-thick gold film on a $SiO_2$ substrate. The real and imaginary dielectric constants of gold were taken from Johnson and Christy's experimental study[39] and fitted with the Drude model to conduct FDTD simulations. The refractive index of glass was set to 1.45. To ensure the precision of simulation, we defined the fine meshes around the nano-slits ($\Delta x = 1$ nm, $\Delta y = 1$ nm and $\Delta z = 5$ nm), and the far-field illumination was set normal to the surface of the sample. To extract multiple hidden eigenmodes under normal-incident pumping conditions, we intentionally restricted the field symmetry conditions in FDTD simulation along the $xz$- and $yz$-planes at the center of the nano-slits (Fig. 4a) and controlled the polarization direction of the excitation source to $x$- or $y$-direction.

Figure 4b shows the maximum $|E|^2$ spectra of the double nano-slits for three representative symmetry conditions with two different polarization directions. Here, the notation of "(odd, even), $x$-pol", for example, indicates that the simulation had an odd and even mirror symmetry of electric fields with respect to the $yz$- and $xz$-plane, respectively, and the excitation source was set to $x$-polarization. The maximum $|E|^2$ was detected above the upper surface of the sample over an area of $10 \times 10$ $\mu m^2$ around the nano-slits. Notably, the six prominent individual eigenmodes ($TE_{00}$, $TE_{01}$, $TE_{02}$, $TE_{10}$, $TE_{03}$, and $TE_{11}$) were observed within a spectral window of 450–800 nm, and their spectra overlapped with respect to one another. This is mainly due to the small

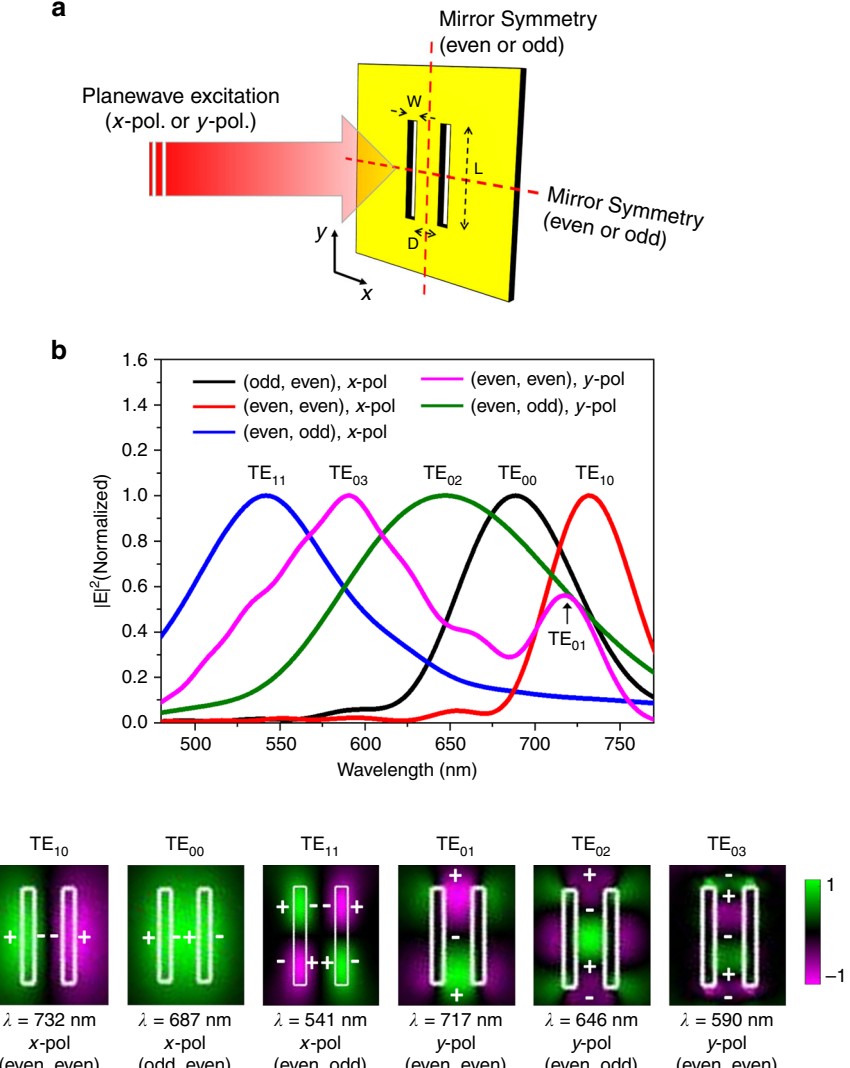

**Fig. 4 Numerical mode analysis of the double nano-slits. a** Schematic sample geometry along with far-field excitation geometry. The polarization direction of the excitation source is selected to either $x$- or $y$-direction. Geometric parameters were taken from the experimentally fabricated double nano-slits ($L = 160$ nm, $W = 20$ nm, and $D = 50$ nm). **b** Maximum $|E|^2$ spectra measured above the upper surface of the sample under mirror symmetry conditions of electric field along the $xz$- and $yz$-plane at the center of the nano-slits. Plots were taken for three representative symmetry conditions with two different polarization directions. Here, the notation of "(odd, even), $x$-pol", for example, indicates that the simulation had odd and even mirror symmetry of electric field with respect to the $yz$- and $xz$-plane, respectively, and the excitation source was set to $x$-polarization. **c** Spatial electric field ($x$ or $y$) maps of various near-field eigenmodes $\mathrm{TE}_{nm}$ obtained 20 nm above the gold surface, identified from Fig. 4b. The maps are displayed in descending order according to the resonance wavelength for each polarization. Plus and minus signs indicate the charge distributions of the antenna mode.

physical dimension of the nano-slits, where the $Q$ factor of each mode was extremely low ($Q < 10$) and the bandwidth of each mode was around 100 nm in wavelength. Consequently, multiple modes could contribute simultaneously to the measured near-field signals at a given excitation wavelength[37]. It is worth noting that the far-field excitation is favorably coupled to lower-order modes as their spatial field is better overlapped with the far-field excitation than the higher-order modes at the input plane. For this reason, the near-field maps obtained in experiment (Fig. 1c) showed mainly the fundamental symmetric mode with minimal signatures of high-order eigenmodes.

For each eigenmode, we obtained its near-field complex field map, which is shown in Fig. 4c in descending order according to the resonance wavelength for each polarization. Here, the field profiles were taken 20 nm above the upper surface of the sample. The polarization state of each mode is indicated as "$x$-pol" or "$y$-

pol" in each figure panel. The $\mathrm{TE}_{00}$ and $\mathrm{TE}_{10}$ modes show an odd and even mirror symmetry with respect to the $yz$-plane, respectively, where the $x$-components of the electric field are dominant. Therefore, they have an even and odd phase symmetry with respect to the $yz$-plane, respectively. In addition, the $\mathrm{TE}_{01}$, $\mathrm{TE}_{02}$, and $\mathrm{TE}_{03}$ correspond to high-order eigenmodes with polarization parallel to the long axis of the nano-slits. The $\mathrm{TE}_{11}$ mode shows a quadrupole mode profile with polarization orthogonal to the long axis of the nano-slits. Plasmonic modes are often subtle to explain due to their short lifetime and low $Q$ factor[40]. In our study, these observed eigenmodes are related to the antenna modes of the double nano-slits whose charge distributions are indicated in Fig. 4c. We found that these mode profiles are in excellent agreement with those obtained in the experiment shown in Fig. 2c. This result confirmed the validity of our experimental eigenmode mapping method.

## Discussion

In summary, we developed a near-field transmission matrix microscopy that can extract the eigenmodes of subwavelength nanostructures whose spectra are too overlapped to selectively excite individual modes by the choice of an excitation wavelength. Under this condition, conventional NSOM can visualize only the dominant modes as it detects the integral of signals from all the contributing modes. In our study, we exploited a new degree of freedom, which is the angle of far-field excitation, to disentangle multiple superposed modes. We experimentally measured the complex-field maps of near-field waves for various angles of far-field excitation and constructed a fully phase-referenced FNTM. This is in contrast with the conventional NSOM, where only a single spatial mode of far-field excitation (normal or a focused illumination) is used. By performing the SVD of the measured matrix, we extracted individual orthogonal eigenmodes. We could visualize the antisymmetric mode, quadruple mode, and other high-order modes of the double nano-slits, which were completely hidden under the conventional NSOM images, and quantify their relative coupling efficiency from their eigenvalues. An analytic model for explaining the working mechanism of eigenmode extraction was presented, and numerical mode analysis confirmed the experimentally observed modes.

The proposed approach is an important advancement in the context of near-field imaging. It provides a new dimension of information that is inaccessible to conventional NSOM, which is the decomposition of superposed modes. In fact, apertureless NSOM can also map the near-field modes of nanostructures[14,41,42]. Cathodoluminescence[43,44], photoemission electron microscopy[45,46], and electron-energy-loss spectroscopy[47,48] that are based on electron microscopy are also useful tools to map local optical modes with high spatial resolution. These methods could often visualize dark modes having no net dipole moments[41,49] as well as bright modes. However, these other methods usually map the dominant modes at the given excitation/detection wavelength. The reason these methods could map the dark modes is because the spectral bandwidths of the dark modes are well separate from those of bright modes. On the contrary, our method can separate multiple spectrally and spatially superposed hybridized bright/dark modes by means of the SVD. This capability can be especially useful in interrogating small-scale nanostructures where the modes of basic building blocks are spectrally so broad that the hybridized modes can have spectral overlaps. The double nano-slits used in our study is one of the good examples.

Our method can potentially be combined with other existing NSOM modalities[14,19,50–53], particularly those based on interferometric detection, and help extract near-field eigenmodes information of various quantities such as electric near-field vector components, magnetic near-field, and time/frequency-resolved measurements[19,54]. The ability to extract the spectrally and spatially overlapped eigenmodes can help designing the nanophotonic devices containing multiple hybridized modes[55,56] and providing new insights into the development of novel nanoscale photonic devices[11,57]. In the present study, the mode decomposition experiment was conducted only for single wavelength excitation. Repeating the same measurements with the scanning of the excitation wavelength could allow the full mapping of the spectra of individual resonance modes constituting the total near-field scattering spectrum. Another noteworthy observation in this study was that higher-order eigenmodes exhibited multiple sub-aperture nodes due to the destructive interference of local near-field waves. This opened a new possibility to resolve the fine structural details of nano-slits whose gap is smaller than the physical size of the probe aperture[58,59] (see Supplementary Note 1). Although this type of resolution enhancement is applicable in limited cases where there are distinct resonance modes, it is still valuable considering that it can retrieve deep subwavelength structural information hidden with conventional NSOM. Considering that the steep reduction in near-field collection efficiency accompanied by the use of the smaller probe sets the practical limit of the spatial resolving power, the capability of imaging with the same resolution by the use of a larger probe aperture will push the ultimate limit of the resolving power in the near-field imaging.

## Methods

**Experimental setup**. To perform the experimental mapping of near-field modes, we integrated a far-field phase modulation system into an NSOM system (Nanonics MV2000) (see Supplementary Note 1 for the detailed experimental setup). The output beam of a laser diode (Thorlabs Inc., LP637-SF70) was enlarged and collimated to uniformly illuminate the SLM (Hamamatsu LCOS-SLM X10468). To measure both the amplitude and phase of the near-field waves on the surface of the nano-slits with their phases fully referenced, we employed a self-interference measurement system without an additional setup for the reference beam line. For this purpose, the SLM generated both the sample and reference beams whose relative phase was controlled in multiples of $\pi/2$. They were demagnified and delivered to the bottom of the NSOM sample stage using an objective lens (Nikon ELWD 40×, NA: 0.6). The overall magnification from the SLM plane to the sample stage was 1/1000×. Near-field waves generated on the upper surface of the nanostructure were measured by scanning the NSOM fiber probe. The aperture diameter of the NSOM probe was 150 or 100 nm. The near-field light captured by the NSOM probe was delivered to a photomultiplier tube (Hamamatsu, H8259-01).

**Simple double-slit model**. The identification of the antisymmetric mode by SVD of the measured FNTM can be understood by a simple double-slit model. The transmission of the incident field through a simple double-slit system can be described by

$$\begin{pmatrix} E_{\text{out}}^1 \\ E_{\text{out}}^2 \end{pmatrix} = \begin{pmatrix} J & K \\ K & J \end{pmatrix} \begin{pmatrix} E_{\text{in}}^1 \\ E_{\text{in}}^2 \end{pmatrix}, \tag{3}$$

where, the vectors $\left(E_{\text{in}}^1, E_{\text{in}}^2\right)$ and $\left(E_{\text{out}}^1, E_{\text{out}}^2\right)$ represent the electric fields at the input and output planes, respectively (the subscripts 1 and 2 indicate the left- and right-hand slits, respectively), $J$ is the coupling constant to the output of the same slit as the input, and $K$ is the coupling to the other slit's output. In the experiment, we cannot directly measure $J$ and $K$ by far-field excitation because the slit separation is too small for far-field illumination to couple light to individual slits. Likewise, the symmetric and antisymmetric modes cannot be individually addressed by the far-field excitation. Instead, the combination of orthogonal modes was measured in the experiment, and SVD was performed to identify the orthogonal modes from the superposed measurements. For the given far-field incident wavevector $\mathbf{k}_{\text{in}}$, the phase difference of the incident wave between the two slits is $\Delta\phi(\mathbf{k}_{\text{in}})$, as defined in the main text. The incident electric field at the two slits can be expressed as $\left(E_{\text{in}}^1, E_{\text{in}}^2\right) = \left(E_0, E_0 e^{-i\Delta\phi(\mathbf{k}_{\text{in}})}\right)$ for any given $\mathbf{k}_{\text{in}}$. We sent 100 different incident wavevectors in the experiment; however, for simplicity, we consider two representative incident wavevectors, one with normal illumination and the other with $\mathbf{k}_{\text{in}}$. The incident electric fields at the two slits are respectively written as $(E_0, E_0)$ and $\left(E_0, E_0 e^{-i\Delta\phi(\mathbf{k}_{\text{in}})}\right)$. Their corresponding output electric fields, which correspond to the quantities measured by the interferometric NSOM in the experiment, can be obtained by inserting these vectors into Eq. (3). Using these two output fields, we can construct a FNTM and analytically calculate the eigenvalues and eigenmodes of $t^\dagger t$ (see Supplementary Note 2 for details). We could verify that the two output eigenvectors are $\mathbf{v}_S \approx \frac{1}{\sqrt{2}}(1, 1)$ (symmetric mode) and $\mathbf{v}_A \approx \frac{1}{\sqrt{2}}(1, -1)$ (antisymmetric mode). Their eigenvalue ratio is calculated as $\frac{\sigma_A}{\sigma_S} = \frac{(1-K/J)^2}{16(1+K/J)^2}\Delta\phi^2$, which is approximately $\frac{\sigma_A}{\sigma_S} \approx \frac{1}{16}\Delta\phi^2$ in the weak-coupling regime.

**Fabrication of nano-slits**. To fabricate the nano-slits with sub-50-nm spacing, we used proximal milling techniques in Ga⁺-based focused ion-beam processes on a sputtered gold film prepared on a silica coverslip (see Supplementary Note 4 for the fabrication details). We intentionally off-designed the milling patterns from the original nano-slit design to make use of the proximity effect of focused ion beam milling. By optimally controlling the distance between rectangular milling patterns and the total milling time, we could realize sub-50-nm spacing between the nano-slits.

## Data availability

The data that support the findings of this study are available from the corresponding author upon reasonable request.

## Code availability

The code used in this study are available from the corresponding author upon reasonable request.

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

## Acknowledgements

This research was supported by IBS-R023-D1. Y.-H. Jin and M.-K. Kim acknowledge support received from the National Research Foundation of Korea (2019M3E4A1078663), the KIST Institutional Program (2E30620-20-051), and the KU-KIST School Project.

## Author contributions

E.S., Q.-H.P., M.-K.K., and Wonshik C. conceived the project. E.S. carried out the measurements using the samples prepared by Y.-H.J and M.-K.K. The experimental data were analyzed by E.S., Y.-H.J., M.-K.K., and Wonshik C. E.S. and Wonshik C. developed the theoretical framework with the help of S.L. and Q.-H.P. E.S., Wonjun C., Y.J., Y.-H.J., and M.-K.K. carried out the numerical simulations to support the experimental results with the help of K.-D.S. and J.A. E.S., Y.-H.J., M.-K.K., and Wonshik C. prepared the paper, and all authors contributed to finalizing the paper.

## Competing interests

The authors declare no competing interests.
