## [Peer Review File · Nature Communications]

Reviewers' comments:

Reviewer #1 (Remarks to the Author):

Near-field transmission microscopy for mapping higher-order eigenmodes of subwavelength nanostructures

by Seo et al.:

The paper presents an experimental method for detection and discrimination of spectrally overlapping plasmonic modes of nanostructures. Two methods are combined: (i) an interferometric reconstruction of the phase and amplitude of electromagnetic field transmitted through a nanostructure, and (ii) post processing of the acquired data using the method of singular-value decomposition (SVD). Both methods have been successfully applied in the field of optics, the present work integrates them in the field of near-field scanning optical microscopy (NSOM) and successfully demonstrates reconstruction of spectrally overlapping near-field modes of a resonant metallic nanostructure.

The method is clearly described in the paper (up to some imperfections mentioned below) and the results are sound and relevant to the nanophotonics community. The originality of the paper lies mainly in the unique combination of the holographic method with the SVD method that is enabled by the spatial-light modulator and the phase stability of the microscope. The method not only allows for identification of spectrally overlapping plasmonic modes (which I think is clearly demonstrated), but it also seems to increase the spatial resolution of the microscope.

I therefore recommend the paper for publication after revision.

Below I list several comments to the authors and questions related to the paper content that should be addressed before the paper can be accepted for publication:

Comment 1: I would like to bring to the author's attention the following papers: Dvorak, P. et al., *Nanoscale*, 10, 21363-21368 (2018), Yang, Y. et al., *ACS Nano*, 13(11), 13595-13601 (2019), where phase and amplitude imaging of plasmonic near-fields using NSOM has been realized in different setups (but the SVD analysis has not been performed).

Comment 2: In the introduction the authors use several vague statements that I believe should be avoided or clarified/concreted/referenced. For example:

- line 33: "coupling with neighboring modes": what are neighboring modes? (in space, frequency?)
- line 44 – 46: "With advancements in fabrication technology, ...": The statement is generic and references are missing.
- line 34 – 35: "This has played a crucial role...": Please, add references or avoid such a generic statement.
- line 71 – 72: "Their optical modes ... are theoretically well understood.": Relevant literature should be cited here to support the statement.
- line 131: polarization was set to "horizontal": "horizontal" is confusing, please clarify (e.g. polarized in the x-direction, as marked in Fig.1b).

Comment 3: I believe that "nano slits" should be used instead of "nano slots" (i.e. the word "slot" should be replaced by "slit" in the paper and supporting material).

Comment 4: Line 154: "...by assigning 100 complex-field maps...": Please, specify whether the maps are taken for 100 different values of k_{in} , it is not clear from the text.

Comment 5: Line 164: eigenvalues of t^+t are defined, but in the caption of Fig. 2 (line 498) the authors talk about “eigenvalues of the transmission matrix” (i.e. of “ t ”). Which quantity is shown in Fig. 2 b,c? Eigenvalues of “ t^+t ” or the singular values of “ t ”? Please, clarify this point. A similar clarification should appear in the section currently entitled “Simple double-slot model”, around line 365.

Comment 6: Line 223 of the supplementary file: please, correct “Error! Reference source not found”.

Q1: The authors exclusively concentrate on NSOM using fiber tips (with a large aperture). Other types of NSOM (such as scattering-type NSOM) are able to reach intrinsic resolution of the order of 10 nm (due to smaller tip size) and acquire both amplitude and phase images of plasmonic near-fields (see e.g. work by Prof. Rainer Hillenbrand). Could the authors discuss the applicability of the method proposed to other types of near-field optical microscopy, e.g. to the scattering-type NSOM?

Q2: In the paper it is assumed that the transmitted reference wave ($E_R(x,y)$) has a flat phase profile (lines 117-118). Could this assumption be verified computationally? The result could be shown in Figure S8 alongside with the other complex field maps.

Q3: The authors interpret the images acquired by NSOM as maps of the near-field transmitted through the nanostructure. However, the large (metal-coated? - details are not specified in the paper) tip of the microscope is held just approximately 1 nm away of the nanostructure surface. Could the authors comment on the influence of the tip on the image formation? How does the tip distort the near-field distribution? It would be useful to demonstrate (or at least discuss/add relevant references about) the general influence of the tip on the image formation.

Q4: The numerical simulations based on the finite-differences-in-time-domain (FDTD) method seem to use (in some configurations) an unphysical constraint on the symmetry of the electromagnetic fields if plane-wave excitation is used (e.g. the (even, even), x-pol configuration). Although it could be argued that the electromagnetic modes generated by such simulations reasonably represent the plasmonic modes, this problem could be circumvented by applying, for example, dipolar excitation (potentially composed of several dipoles) of appropriate symmetry. Do the authors think that this more physical approach could be desirable? Also, which implementation of FDTD has been used?

Q5: How do the authors define plasmonic eigenmodes? The definition of plasmonic eigenmodes is currently under debate (see e.g. work by Prof. Stephen Hughes). Could the experimental images be linked to a more rigorous definition of plasmonic eigenmodes? If yes, this would be a valuable contribution to the field.

Reviewer #2 (Remarks to the Author):

In the paper “Near-field transmission matrix microscopy for mapping high-order eigenmodes of subwavelength nanostructures”, Seo et al. describe a numerical method based on a singular value decomposition (SVD) approach, to map the higher-order eigenmodes of a nanostructure characterized by a near field optical microscope.

The technic is original, elegant and seems to work very nicely. I have no comments on the scientific approach; everything seems correct on the analysis side. As well, the detailed information would allow

researchers to reproduce this technic. Though I wonder what is the influence of the near field tip on this technic, since the distance tip-sample is about 1nm, I'd think the coupling between them must be quite strong and therefore affect the results. No mention of this interaction is made.

But my concern is more towards the applicability of the method, since, from what I understand, the approach is only useful for specific nanostructures, like the one used in this paper. Indeed, to acquire data over the dark modes of a photonic nanostructure, these modes need to be excited by a far-field illumination, which by definition is not the case in many nanostructures. Therefore, I wonder about the interest of this work to the broad community of Nature Communications.

My recommendation would be to send it to a more specialized journal, such as Nanoletters, ACS nano, Nanophotonics or ACS photonics.

Reviewer #3 (Remarks to the Author):

In this work, Seo et al. use the singular value decomposition in conjunction with aperture NSOM measurements to map out the higher order mode of nanoparticles. This is an interesting idea, and certainly extends the functionality of aperture NSOM's beyond the standard application to guided mode fields. This is in contrast to the authors' motivation, which stems from an understanding of the hybridization that could arise from the integration of many photonic elements on a single chip. To be clear, I do not think that the authors have shown anything to do with the mapping or understanding of any such potential hybridization. To me, however, the extension of the NSOM capabilities is sufficient to warrant publication in Nature Communication (at least, once the authors have dealt with several issues that I outline below). I could, however, fully support an editorial decision that given the lack of connection to large-scale photonics, this work can be published in a Journal such as Nano Letters or ACS Photonics.

1. The authors spend most of their introduction/motivation describing the hybridization of modes on large-scale and complex photonic chips, arguing that a method is required to measure these modes. In fact, most integrated photonic circuits are designed exactly to avoid (or at least minimize) this problem, in essence to avoid cross-talk between components as that limits the fidelity of the circuit. It is therefore unclear what hybridization the authors talk about, in this context. Rather, what the authors seem to discuss are the higher-order modes that many nano-particles, or composite nanoparticles (such as dimers) support. One does not have anything to do with the other. Therefore, for this work to make sense, the authors should motivate their work in terms of the problem that they actually address, and show why this is important enough to warrant publication in Nature Communications.

2. What attracts me to this work is that the authors use an aperture NSOM to study the modes of nanoparticles, whereas this type of NSOM is typically used to study guided modes. Consequently, the authors should contrast their results with methods that routinely image modes of nanoparticles, such as apertureless NSOM and cathodeluminescence.

3. As the authors are introducing a new NSOM measurement protocol, the performance of the method is very important. They do a very nice job letting us know how long a complete measurement (with 100 angles) takes, and showing how taking different number of measurements affects the final reconstruction (Fig. 3). What I am missing, in this vein, is a similar discussion on the measurement resolution. In this work, data is taken with 25 nm steps (line 139), which is well below the resolution of the image formation process (see Light, Science and Applications, 8, 28 (2019)). I think a study of

the reconstructed image as a function of the measurement step size is hence very important; it seems likely to me that they are currently over-sampling and therefor needlessly increasing their measurement time. To me, a plot of the reconstruction fidelity of each mode as a function of step-size is missing.

Minor points:

4. Around lines 51-53, the authors argue that conventional NSOM would only map the dominant mode at the excitation wavelength. This argument only holds for nanoparticles. For propagating modes a Fourier Transform can be used to separate the different modes/harmonics.

5. On line 72, the authors state that the properties of nano-slots are well understood. This statement begs for references.

6. On line 117, the authors state that the phase of the reference field is taken to be constant. But the authors measure across different resonances, each which will contribute a different phase spectrum. Or, do they mean constant in space, but not in frequency? In either case, this should be clarified.

7. On line 140, the authors state that they hold the tip about 1 nm away from the sample. This is much closer than the 10-25 nm typically reported. How do the authors know that they are at this distance and how do they manage to scan at this height?

8. I read the statement on line 152-3 that the image depends on $k^{\wedge}in$ because it depends on $k^{\wedge}in$, which is meaningless. Please rewrite so that this means something.

We thank the reviewers for their constructive and critical comments, which helped us to improve the quality of our manuscript. In our revision, we addressed each and every issue raised by the reviewers. In particular, we clarified that the main motivation of our study is to provide a near-field imaging tool that can experimentally elucidate the way modes of the subwavelength nanostructures are hybridized. We also provided the rationale and related references supporting the applicability and implication of our method in designing and fabricating complex nanophotonic devices. Furthermore, the comparison with other mode-mapping methods including apertureless NSOM and cathodoluminescence was discussed. We emphasized the unique benefit of our method in mapping the spectrally and spatially overlapped modes along with the innate limitations. Technical issues such as the effect of the tip-sample interaction and the optimal NSOM probe scanning step were all addressed by conducting additional analysis. All the changes were highlighted in red in the revised manuscript and supplementary information.

Reviewer #1

The paper presents an experimental method for detection and discrimination of spectrally overlapping plasmonic modes of nanostructures. Two methods are combined: (i) an interferometric reconstruction of the phase and amplitude of electromagnetic field transmitted through a nanostructure, and (ii) post processing of the acquired data using the method of singular-value decomposition (SVD). Both methods have been successfully applied in the field of optics, the present work integrates them in the field of near-field scanning optical microscopy (NSOM) and successfully demonstrates reconstruction of spectrally overlapping near-field modes of a resonant metallic nanostructure.

The method is clearly described in the paper (up to some imperfections mentioned below) and the results are sound and relevant to the nanophotonics community. The originality of the paper lies mainly in the unique combination of the holographic method with the SVD method that is enabled by the spatial-light modulator and the phase stability of the microscope. The method not only allows for identification of spectrally overlapping plasmonic modes (which I think is clearly demonstrated), but it also seems to increase the spatial resolution of the microscope.

I therefore recommend the paper for publication after revision.

Below I list several comments to the authors and questions related to the paper content that should be addressed before the paper can be accepted for publication.

We appreciate the reviewer's acknowledging the originality of our work and the key benefits – the identification of the spectrally overlapping plasmonic modes and the potential increase of effective spatial resolving power. In the following, we addressed all the reviewer's valuable comments and suggestions.

*Comment 1: I would like to bring to the author's attention the following papers: Dvorak, P. et al., *Nanoscale*, 10, 21363-21368 (2018), Yang, Y. et al., *ACS Nano*, 13(11), 13595-13601 (2019), where phase and amplitude imaging of plasmonic near-fields using NSOM has been realized in different setups (but the SVD analysis has not been performed).*

We added the suggested references to the revised manuscript and inserted the following sentence in the main text.

“Various phase imaging NSOM techniques have been developed in the past¹⁻⁴, but here we made an important addition.”

1. Feber, B. le, Rotenberg, N., Beggs, D. M. & Kuipers, L. Simultaneous measurement of nanoscale electric and magnetic optical fields. *Nat Photonics* **8**, 43–46 (2013)
2. Schnell, M., Carney, P. S. & Hillenbrand, R. Synthetic optical holography for rapid nanoimaging. *Nat Commun* **5**, 3499 (2014)
3. Dvorak, P. et al. Near-field digital holography: a tool for plasmon phase imaging. *Nanoscale* **10**, 21363-21368 (2018)
4. Yang, Y. T., Zhai, C. H., Zeng, Q., Khan, A. & Yu, H. Quantitative Amplitude and Phase Imaging with Interferometric Plasmonic Microscopy. *Acs Nano* **13**, 13595-13601 (2019)

Comment 2: In the introduction the authors use several vague statements that I believe should be avoided or clarified/concreted/referenced. For example:

- line 33: “coupling with neighboring modes”: what are neighboring modes? (in space, frequency?)

We intended to describe mode coupling among spatially and spectrally overlapped modes, but the term neighboring mode was vague. We clarified the sentences as follows and added relevant references to support them.

“NSOM has served as a general and robust tool to map the local near-field for unveiling the underlying process of nanoscopic physical phenomena and optimizing the performance of nanophotonic devices¹⁻⁷. It has been especially useful to visualize the formation of these optical modes and their coupling with the other spatially and spectrally overlapped modes²⁻⁶.”

1. Rotenberg, N. & Kuipers, L. Mapping nanoscale light fields. *Nature Photonics* **8**, 919-926 (2014).
2. Xavier, J., Vincent, S., Meder, F. & Vollmer, F. Advances in optoplasmonic sensors - combining optical nano/microcavities and photonic crystals with plasmonic nanostructures and nanoparticles. *Nanophotonics* **7**, 1-38 (2018).
3. Tali, S. A. S. & Zhou, W. Multiresonant plasmonics with spatial mode overlap: overview and outlook. *Nanophotonics* **8**, 1199-1225 (2019).
4. Hecht, B. et al. Scanning near-field optical microscopy with aperture probes: Fundamentals and applications. *Journal of Chemical Physics* **112**, 7761-7774 (2000).
5. Davis, T. J., Gomez, D. E. & Vernon, K. C. Simple Model for the Hybridization of Surface Plasmon Resonances in Metallic Nanoparticles. *Nano Letters* **10**, 2618-2625 (2010).
6. Lovera, A., Gallinet, B., Nordlander, P. & Martin, O. J. F. Mechanisms of Fano Resonances in Coupled Plasmonic Systems. *Acs Nano* **7**, 4527-4536 (2013).

- line 44 – 46: “With advancements in fabrication technology, ...”: The statement is generic and references are missing.

- line 34 – 35: “This has played a crucial role...”: Please, add references or avoid such a generic statement.

We added the following review papers to support the statements.

1. Rotenberg, N. & Kuipers, L. Mapping nanoscale light fields. **8**, 919–926 (2014).
2. Xavier, J., Vincent, S., Meder, F. & Vollmer, F. Advances in optoplasmonic sensors – combining optical nano/microcavities and photonic crystals with plasmonic nanostructures and nanoparticles. *Nanophotonics* **7**, 1–38 (2018)

- line 71 – 72: “Their optical modes ... are theoretically well understood.”: Relevant literature should be cited here to support the statement.

We added the following references elucidating the formation of hybridized eigenmodes by the mode coupling between the nano-slits.

1. Park, Y., Kim, J., Roh, Y.-G. & Park, Q.-H. Optical slot antennas and their applications to photonic devices. *Nanophotonics* **7**, 1617–1636 (2018)
2. Lee, S., Park, Y., Kim, J., Roh, Y.-G. & Park, Q.-H. Selective bright and dark mode excitation in coupled nanoantennas. *Opt Express* **26**, 21537–21545 (2018)

- line 131: polarization was set to “horizontal”: “horizontal” is confusing, please clarify (e.g. polarized in the x-direction, as marked in Fig.1b).

Following the suggestion, we modified ‘horizontal’ to ‘x-direction.’

Comment 3: I believe that “nano slits” should be used instead of “nano slots” (i.e. the word “slot” should be replaced by “slit” in the paper and supporting material).

The word ‘slot’ was all replaced by ‘slit’ in the revised manuscript.

Comment 4: Line 154: “...by assigning 100 complex-field maps...”: Please, specify whether the maps are taken for 100 different values of k_{in} , it is not clear from the text.

We revised the phrase to make it clear that 100 complex-field maps were taken by different values of \vec{k}^{in} .

Comment 5: Line 164: eigenvalues of $t^{\dagger}t$ are defined, but in the caption of Fig. 2 (line 498) the authors talk about “eigenvalues of the transmission matrix” (i.e. of “ t ”). Which quantity is shown in Fig. 2 b,c? Eigenvalues of “ $t^{\dagger}t$ ” or the singular values of “ t ”? Please, clarify this point. A similar clarification should appear in the section currently entitled “Simple double-slot model”, around line 365.

When we refer to the eigenvalues, they meant the eigenvalues of $t^{\dagger}t$. We revised the phrases to clarify this definition.

Comment 6: Line 223 of the supplementary file: please, correct “Error! Reference source not found”.

Corrected.

Q1: The authors exclusively concentrate on NSOM using fiber tips (with a large aperture). Other types of NSOM (such as scattering-type NSOM) are able to reach intrinsic resolution of the order of 10 nm (due to smaller tip size) and acquire both amplitude and phase images of plasmonic near-fields (see e.g. work by Prof. Rainer Hillenbrand). Could the authors discuss the applicability of the method proposed to other types of near-field optical microscopy, e.g. to the scattering-type NSOM?

In the Discussion section of the original manuscript, we briefly discussed the possibility of applying our method to other NSOM modalities including the scattering-type NSOM (see the sentence below). The proposed mode mapping method by the singular value decomposition of FNTM can be extended to any types of NSOM as long as they record the phase and amplitude of near-field waves.

“Our method can potentially be combined with other existing NSOM modalities^{12,33-36}, particularly those based on interferometric detection, and help extract near-field eigenmodes information of various quantities such as electric near-field vector components, magnetic near-field, and time/frequency-resolved measurements^{36,37}.”

In the revision, we added the following paper from Hillenbrand group recording the phase and amplitude using scattering-type NSOM.

1. Schnell, M., Carney, P. S. & Hillenbrand, R. Synthetic optical holography for rapid nanoimaging. *Nat. Commun.* **5**, 3499 (2014)

Q2: *In the paper it is assumed that the transmitted reference wave ($E_R(x,y)$) has a flat phase profile (lines 117-118). Could this assumption be verified computationally? The result could be shown in Figure S8 alongside with the other complex field maps.*

We appreciate the reviewer's suggestion. Our FDTD simulation results in Fig. R1 show that the near-field phase is spatially flat across the double nano-slits for the normally incident reference wave. We added this result to Fig. S10 and referred this addition to in the main text.

Figure R1. Electric field distribution for the normal illumination of the reference wave. **a** and **b**, Normalized amplitude map and the phase map of the near-field wave at the surface of the sample, respectively. **c**, Line profiles of amplitude and phase along the transverse direction of double nano-slits indicated as white dashed line in **a**. Scale bar, 200 nm.

Q3: *The authors interpret the images acquired by NSOM as maps of the near-field transmitted through the nanostructure. However, the large (metal-coated? - details are not specified in the paper) tip of the microscope is held just approximately 1 nm away of the nanostructure surface. Could the authors comment on the influence of the tip on the image formation? How does the tip distort the near-field distribution? It would be useful to demonstrate (or at least discuss/add relevant references about) the general influence of the tip on the image formation.*

There was a mistake in translating manufacturer's specification. In fact, the distance between the tip and sample was maintained to be 20 nm, not 1 nm. We corrected this error in the revised manuscript. Our probe is made of an Au-Cr coated tapered waveguide, and this was added to the revised manuscript.

We verified that the tip-sample interaction in our experiment was negligible from the good agreements between the experimentally identified eigenmodes with those acquired by the FDTD

simulations (Fig. S5). In this response letter, we conducted additional FDTD simulations and confirmed that the effect of tip-sample interaction is negligible for the tip-sample distance of 20 nm, while the modification of the near-field is significant for the tip-sample distance of 1 nm (Fig. R2).

The layout of the sample geometry for the additional FDTD simulation is shown in Fig. R2a. The NSOM aperture tip was modeled as a PEC-glass-PEC slab. We considered two cases of the tip-sample distance z_t , i.e. $z_t = 1$ nm and 20 nm. And the relative lateral position of the tip to the double nano-slits was varied. Figures R2b-d show the electric field at the sample surface (red dashed line in Fig. R2a) when the center of the metal coating (x_1), the fiber-metal boundary of aperture (x_2), and the center of the aperture (x_3) are matched to the center of the sample (x_0), respectively. For comparison, the near-field map at the sample surface is shown in the absence of the tip (black dashed curves).

The near-field was disturbed by the tip when the tip-sample distance is 1 nm for all three cases. In particular, the disturbance was pronounced when the metal side of the tip was positioned above nano-slits (Figs. R2b and c). On the contrary, when the tip-sample distance is 20 nm, the near-field maintains a similar shape to that without the tip for all the cases. These simulation results support that the tip-sample interaction was not significant in our experiments.

Figure R2. The effect of the tip-sample interaction. **a**, Simulation layout for various tip-sample geometries. **b-d**, The E-field profile on gold surface when the tip is either 1 nm (red curve) or 20 nm (blue curve) above the sample for the cases when the center of the metal coating (x_1), the fiber-metal boundary of aperture (x_2), and the center of the aperture (x_3), are matched to the center of the double nano-slits (x_0), respectively. Black dashed curves are the near-field profiles in the absence of the tip.

Q4: *The numerical simulations based on the finite-differences-in-time-domain (FDTD) method seem to use (in some configurations) an unphysical constraint on the symmetry of the electromagnetic fields if plane-wave excitation is used (e.g. the (even, even), x-pol configuration).*

Although it could be argued that the electromagnetic modes generated by such simulations reasonably represent the plasmonic modes, this problem could be circumvented by applying, for example, dipolar excitation (potentially composed of several dipoles) of appropriate symmetry. Do the authors think that this more physical approach could be desirable? Also, which implementation of FDTD has been used?

We thank the reviewer for pointing this out. We agree with the reviewer that our numerical simulation based on the FDTD method used rather unphysical constraint on the symmetry of the electromagnetic fields of plane wave source to find the hidden eigenmodes of lossy plasmonic cavity. As suggested by the reviewer, the cavity eigenmodes can be found by using a dipole excitation and analyzing the electromagnetic fields remaining in the cavity after the source is turned off. However, since the plasmonic cavity (or plasmonic slit antenna) formed by the size of nano-slits used in our study is very lossy, the lifetime of the eigenmodes is typically less than 3.5 fs or the Q-factor is less than 10. This makes it extremely difficult to analyze the electromagnetic fields remaining in the cavity.

In order to resolve this issue, we measured the energy stored in the plasmonic cavity under the continuous plane wave pumping condition with constraints on the symmetry of electromagnetic fields. Although this source appears to impose unphysical constraints, it can be understood as imposing a very high-resolution beam shaping to the double nano-slit cavity to excite the specific eigenmode, as shown in Fig. R3. Under these pumping conditions, the accumulation of energy in the cavity is due to the presence of eigenmode in specific phase conditions (or symmetry conditions). Therefore, this method finds and separates multiple hidden eigenmodes present in the highly lossy plasmonic cavity.

* Example:

$$(\text{even}, \text{odd}) \leftrightarrow (\varphi_1, \varphi_2, \varphi_3, \varphi_4) = (0, 0, \pi, \pi)$$

$$(\text{odd}, \text{odd}) \leftrightarrow (\varphi_1, \varphi_2, \varphi_3, \varphi_4) = (0, \pi, \pi, 0)$$

Figure R3. Computation using symmetry constraints are similar to calculation using high-resolution beam shaping.

Based on the above reasons, we used the following FDTD method:

- 1) First, we applied symmetry conditions for the xz - and yz -planes at $x=0$ and $y=0$ in the FDTD simulation domain and applied open boundary conditions to all simulation boundaries.
- 2) Then, we generated a plane wave source 1 μm above the surface of the double nano-slits and measured the spectrum of the maximum $|E|^2$ in the double nano-slits.

We think this calculation method is an effective way to find the hidden eigenmodes of the highly lossy plasmonic cavity. The validity of this computation method is confirmed from the fact that the mode profiles of the simulation are quite consistent with the experimental results.

Q5: How do the authors define plasmonic eigenmodes? The definition of plasmonic eigenmodes is currently under debate (see e.g. work by Prof. Stephen Hughes). Could the experimental images be linked to a more rigorous definition of plasmonic eigenmodes? If yes, this would be a valuable contribution to the field.

Since the plasmon cavity modes typically have a short lifetime, it is difficult to define the eigenmodes as mentioned by the reviewer. In this respect, small plasmonic cavities with high radiation losses are often interpreted as optical antennas. The antenna mode, such as a dipole or a quadrupole antenna

mode, is generally defined by the way the charges are distributed in the conductor. Therefore, one can define the eigenmodes of lossy plasmonic cavity based on the symmetry of charge distribution. The six eigenmodes obtained in the experiment (Fig. 2) and FDTD simulations (Fig. 4c) can be interpreted as the radiation from antenna modes whose charge distributions are indicated in Fig. R4.

Figure R4. Spatial electric field maps and the charge distributions of the six eigenmodes.

We added the charge distribution in Fig. R4 to Fig. 4c in the revised manuscript and added the following sentence to the revised manuscript.

“Plasmonic modes are often subtle to explain due to their short lifetime and low Q factor¹. In our study, these observed eigenmodes are related to the antenna modes of the double nano-slits whose charge distributions are indicated in Fig. 4c.”

1. Dezfouli, M. K. & Hughes, S. Regularized quasinormal modes for plasmonic resonators and open cavities. *Phys Rev B* **97**, 115302 (2018)

Reviewer #2

In the paper “Near-field transmission matrix microscopy for mapping high-order eigenmodes of subwavelength nanostructures”, Seo et al. describe a numerical method based on a singular value decomposition (SVD) approach, to map the higher-order eigenmodes of a nanostructure characterized by a near field optical microscope.

The technic is original, elegant and seems to work very nicely. I have no comments on the scientific approach; everything seems correct on the analysis side. As well, the detailed information would allow researchers to reproduce this technic. Though I wonder what is the influence of the near field tip on this technic, since the distance tip-sample is about 1nm, I’d think the coupling between them must be quite strong and therefore affect the results. No mention of this interaction is made.

We deeply appreciate the reviewer’s acknowledging the originality of our method and the elegance of our experimental demonstration. In the revised manuscript, we addressed all the reviewers’ critical comments and valuable suggestions to make our manuscript as clear as possible.

Regarding the tip-sample distance, we made a mistake in translating the manufacturer’s specifications. In fact, the tip-sample distance was 20 nm, not 1 nm.

We verified that the tip-sample interaction in our experiment was negligible from the good agreements between the experimentally identified eigenmodes with those acquired by the FDTD simulations (Fig. R5). In this response letter, we conducted additional FDTD simulations and confirmed that the effect of tip-sample interaction is negligible for the tip-sample distance of 20 nm, while the modification of the near-field is significant for the tip-sample distance of 1 nm (Fig. R5).

The layout of the sample geometry in the additional FDTD simulation is shown in Fig. R5a. The NSOM aperture tip was modeled as a PEC-glass-PEC slab. We considered two cases of the tip-sample distance z_t , i.e. $z_t = 1$ nm and 20 nm. And the relative lateral position of the tip to the double nano-slits was varied. Figures R5b-d show the electric field at the sample surface (red dashed line in Fig. R5a) when the center of the metal coating (x_1), the fiber-metal boundary of aperture (x_2), and the

center of the aperture (x_3) are matched to the center of the sample (x_0), respectively. For comparison, the near-field map at the sample surface is shown in the absence of the tip (black dashed curves).

The near-field was disturbed by the tip when the tip-sample distance is 1 nm for all three cases. In particular, the disturbance was pronounced when the metal side of the tip was positioned above nano-slits (Figs. R5b and c). On the contrary, when the tip-sample distance is 20 nm, the near-field maintains a similar shape to that without the tip for all the cases. These simulation results support that the tip-sample interaction was not significant in our experiments.

Figure R5. The effect of tip-sample interaction. **a**, Simulation layout for various tip-sample geometries. **b-d**, The E-field profile on gold surface when tip is either 1 nm (red curve) or 20 nm (blue curve) above the sample for the cases when the center of the metal coating (x_1), the fiber-metal boundary of aperture (x_2), and the center of the aperture (x_3), are matched to the center of the sample (x_0), respectively. Black dashed curves are the near-field profiles in the absence of the tip.

But my concern is more towards the applicability of the method, since, from what I understand, the approach is only useful for specific nanostructures, like the one used in this paper. Indeed, to acquire data over the dark modes of a photonic nanostructure, these modes need to be excited by a far-field illumination, which by definition is not the case in many nanostructures. Therefore, I wonder about the interest of this work to the broad community of Nature Communications.

We would like to emphasize that the main theme of our work is to provide an experimental tool to investigate the way modes of nanostructures are hybridized, especially when they are spectrally and spatially superposed. Double nano-slits were used as an exemplary platform to demonstrate the benefit of the proposed method. The extraction of dark modes in the double nano-slits shows the extreme capability of our method. By the recording of the far- to near-field transmission matrix, we could extract those high-order modes that can barely be excited by the far-field illumination. We

believe that our method could be useful in the physical implementation of the complex photonic devices. The detailed mode mapping capability can play a crucial role when nanostructures are made smaller because the fabrication is more likely to be off from the design and the modes of the building blocks are spectrally more broadened. As such, we believe that the proposed method is appealing to not only the NSOM community, but also the nanophotonic device community in general.

We added the following sentence to the Discussion section to introduce the potential application of the proposed method.

“The ability to extract the spectrally and spatially overlapped eigenmodes can help designing the nanophotonic devices containing multiple hybridized modes^{1,2} and providing new insights into the development of novel nanoscale photonic devices^{3,4}.”

1. Yin, L. Y., Huang, Y. H., Wang, X., Ning, S. T. & Liu, S. D. Double Fano resonances in nanoring cavity dimers: The effect of plasmon hybridization between dark subradiant modes. *Aip Advances* **4** (2014)
2. Guo, H. C. et al. Resonance hybridization in double split-ring resonator metamaterials. *Optics Express* **15**, 12095-12101 (2007)
3. Xavier, J., Vincent, S., Meder, F. & Vollmer, F. Advances in optoplasmonic sensors – combining optical nano/microcavities and photonic crystals with plasmonic nanostructures and nanoparticles. *Nanophotonics* **7**, 1–38 (2018)
4. Yan, Z. D., Qian, L. N., Zhan, P. & Wang, Z. L. Generation of tunable double Fano resonances by plasmon hybridization in graphene-metal metamaterial. *Applied Physics Express* **11** (2018)

Reviewer #3

In this work, Seo et al. use the singular value decomposition in conjunction with aperture NSOM measurements to map out the higher order mode of nanoparticles. This is an interesting idea, and certainly extends the functionality of aperture NSOM's beyond the standard application to guided mode fields. This is in contrast to the authors' motivation, which stems from an understanding of the hybridization that could arise from the integration of many photonic elements on a single chip. To be clear, I do not think that the authors have shown anything to do with the mapping or understanding of any such potential hybridization. To me, however, the extension of the NSOM capabilities is sufficient to warrant publication in Nature Communication (at least, once the authors have dealt with several issues that I outline below). I could, however, fully support an editorial decision that given the lack of connection to large-scale photonics, this work can be published in a Journal such as Nano Letters or ACS Photonics.

We are thankful to the reviewer for acknowledging that the extension of the NSOM capabilities that our method provides is sufficient to warrant the publication in Nature Communications. In our work, we demonstrated the detailed mapping of the hybridized modes of double nano-slits formed by the coupling between the spectrally and spatially overlapped modes of individual nano-slits. In particular, we could map the higher order dark modes that are inaccessible to the conventional aperture-type NSOM. Although the nano-slits are not large-scale photonic devices, our method elucidating how the modes of the fundamental building blocks such as nano-slits are hybridized will certainly be useful. We believe that this could potentially be helpful in designing and fabricating complex photonic devices, as have been the case with the other novel NSOM techniques.

We think that the reviewer's reservation is due to some of the claims in the introduction, which are inconsistent with what was demonstrated. We revised the manuscript to best accommodate the reviewer's suggestion given in the comment #1 below and clarify the motivation of our work in such a way to address the problem that we solved.

1. The authors spend most of their introduction/motivation describing the hybridization of modes on large-scale and complex photonic chips, arguing that a method is required to measure these modes. In fact, most integrated photonic circuits are designed exactly to avoid (or at least minimize) this problem, in essence to avoid cross-talk between components as that limits the fidelity of the circuit. It is therefore unclear what hybridization the authors talk about, in this context. Rather, what the authors seem to discuss are the higher-order modes that many nano-particles, or composite nanoparticles (such as dimers) support. One does not have anything to do with the other. Therefore, for this work to make sense, the authors should motivate their work in terms of the problem that they actually address, and show why this is important enough to warrant publication in Nature Communications.

The reviewer's insightful comments helped us to clarify the main motivation of our study. In developing integrated nanophotonic devices having multiple functionalities, it is often necessary to design the hybridized modes having distinct resonances with minimal cross-talks, as the reviewer remarked, by properly engineering the coupling among the spectrally and spatially overlapped modes of the constituting building blocks. The main motivation of our study is to provide an experimental tool that can elucidate the way these hybridized modes are formed in the physical implementation of the integrated devices. This will be more crucial when nanostructures are made smaller because the fabrication is more likely to be off from the design and the modes of the building blocks are spectrally more broadened.

In our original manuscript, we stated as if the integrated nanophotonic devices will have multiple spectrally overlapped higher order modes and our main goal is to map those higher order modes. As the reviewer pointed out, this is not the usual design goal. We revised the introduction of the manuscript as follows such that our study is to provide an experimental tool to investigate the mode hybridization.

“With advancements in fabrication technology, there have been growing interest in fabricating smaller nanostructures and integrating them at a greater density with a view to merging multiple functionalities within a smaller chip^{1,2}. In this process, it is often necessary to design the hybridized modes having distinct resonances with minimal cross-talks by properly engineering the coupling among the spectrally and spatially overlapped modes of the constituent nanostructures³⁻⁵. In the physical implementation of the designed nanostructures, it will be important to have experimental tools that can visualize the way these hybridized modes are formed. This becomes more crucial when nanostructures are made smaller. Mode hybridization tends to be more complicated because the modes of the basic building blocks such as nanoparticles, nanorods and nano-slits are spectrally more broadened⁶. Furthermore, the fabrication is more likely to be off from the design.

Here, we proposed a near-field imaging method termed a near-field transmission matrix microscopy for simultaneously mapping multiple hybridized eigenmodes that are formed by the coupling between the modes of the constituent nanostructures.”

1. Rotenberg, N. & Kuipers, L. Mapping nanoscale light fields. *Nature Photonics* **8**, 919-926 (2014)
2. Xavier, J., Vincent, S., Meder, F. & Vollmer, F. Advances in optoplasmonic sensors – combining optical nano/microcavities and photonic crystals with plasmonic nanostructures and nanoparticles. *Nanophotonics-berlin* **7**, 1–38 (2018)
3. Tali, S. A. S. & Zhou, W. Multiresonant plasmonics with spatial mode overlap: overview and outlook. *Nanophotonics-berlin* **8**, 1199–1225 (2019)
4. Li, H. A., Tian, X. R., Huang, Y. Z., Fang, L. & Fang, Y. R. Quantitatively analyzing the mechanism of giant circular dichroism in extrinsic plasmonic chiral nanostructures by tracking the interplay of electric and magnetic dipoles. *Nanoscale* **8**, 3720-3728 (2016)
5. Pascale, M., Miano, G., Tricarico, R. & Forestiere, C. Full-wave electromagnetic modes and hybridization in nanoparticle dimers. *Scientific Reports* **9** (2019)

6. Ye, J. et al. Plasmonic Nanoclusters: Near Field Properties of the Fano Resonance Interrogated with SERS. *Nano Letters* **12**, 1660-1667 (2012)

2. What attracts me to this work is that the authors use an aperture NSOM to study the modes of nanoparticles, whereas this type of NSOM is typically used to study guided modes. Consequently, the authors should contrast their results with methods that routinely image modes of nanoparticles, such as apertureless NSOM and cathodoluminescence.

We deeply appreciate the reviewer's suggestion. Apertureless NSOM can map the near-field modes of nanostructures better than typical aperture-type NSOM due to its small tip size and the better proximity of the tip to the sample surface. Cathodoluminescence, photoemission electron microscopy, and electron-energy-loss spectroscopy that are based on electron microscopy are also fascinating tools to map local optical modes with high spatial resolution. These methods could often visualize dark modes having no net dipole moments as well as bright modes.

The aperture-type NSOM detects leaked near-field waves or guided modes at approximately 20-30 nm away from the sample surface. Therefore, it is inevitable to lose fine spatial details whose spatial frequencies exceeding $2k_0$ or $3k_0$ set by the size of the probe aperture. However, the captured guided modes still contain the near-field waves that resemble the original near-field to the extent that the charge distributions of the associated modes can be conjectured (Fig. R7). Our far- to near-field transmission matrix approach is an effective and sensitive method to extract those fine details that are invisible under the conventional aperture-type NSOM. The combination of various angles of oblique far-field illuminations and the azimuthal rotation of the double nano-slits allows us to couple light to both bright and dark modes similar to the apertureless NSOM study. Singular value decomposition could extract weakly coupled dark modes as well as strong bright modes.

Figure R6. Near-field profile depending on the distance from the sample surface. a, Sample geometry and the near-field sampling line (red dashed line) at a distance z_m from the sample surface. **b,** The E-field profile for various values of z_m . Up to the distance of around 20 nm, near-field profiles preserve the local field profile associated with the sample structure.

In comparison with the apertureless NSOM and those methods based on electron microscopy, our method is low in spatial resolving power due to the size of the probe aperture. However, these other methods usually map the dominant modes at the given excitation/detection wavelength. The reason these methods could map the dark modes is because the spectral bandwidths of the dark modes are well separate from those of bright modes. On the contrary, our method can separate multiple spectrally and spatially superposed and hybridized bright/dark modes by means of the singular value decomposition. This capability can be especially useful in interrogating small-scale nanostructures where the modes of basic building blocks are spectrally so broad that the multiple hybridized modes can have spectral overlaps. The double nano-slits used in our study is one of the good examples.

We added the following paragraph to the Discussion section of the revised manuscript together with the related references.

“The proposed approach is an important advancement in the context of near-field imaging. It provides a new dimension of information that is inaccessible to conventional NSOM, which is the decomposition of superposed modes. In fact, apertureless NSOM can also map the near-field modes of nanostructures¹⁻³. Cathodoluminescence^{4,5}, photoemission electron microscopy^{6,7}, and electron-energy-loss spectroscopy^{8,9} that are based on electron microscopy are also useful tools to map local optical modes with high spatial resolution. These methods could often visualize dark modes having no net dipole moments^{10,11} as well as bright modes. However, these other methods usually map the dominant modes at the given excitation/detection wavelength. The reason these methods could map the dark modes is because the spectral bandwidths of the dark modes are well separate from those of bright modes. On the contrary, our method can separate multiple spectrally and spatially superposed hybridized bright/dark modes by means of the singular value decomposition. This capability can be especially useful in interrogating small-scale nanostructures where the modes of basic building blocks are spectrally so broad that the hybridized modes can have spectral overlaps. The double nano-slits used in our study is one of the good examples.”

1. Dorfmueller, J. *et al.* Fabry-Pérot Resonances in One-Dimensional Plasmonic Nanostructures. *Nano Lett* **9**, 2372–2377 (2009)
2. Jones, A. C. *et al.* Mid-IR Plasmonics: Near-Field Imaging of Coherent Plasmon Modes of Silver Nanowires. *Nano Lett* **9**, 2553–2558 (2009)
3. Schnell, M. *et al.* Controlling the near-field oscillations of loaded plasmonic nanoantennas. *Nat Photonics* **3**, 287–291 (2009)
4. Sapienza, R. *et al.* Deep-subwavelength imaging of the modal dispersion of light. *Nat Mater* **11**, 781–787 (2012)
5. Yamamoto, N., Ohtani, S. & Abajo, F. J. G. de. Gap and Mie Plasmons in Individual Silver Nanospheres near a Silver Surface. *Nano Lett* **11**, 91–95 (2011)
6. Spektor, G. *et al.* Revealing the subfemtosecond dynamics of orbital angular momentum in nanoplasmonic vortices. *Science* **355**, 1187–1191 (2017)
7. Kahl, P. *et al.* Direct Observation of Surface Plasmon Polariton Propagation and Interference by Time-Resolved Imaging in Normal-Incidence Two Photon Photoemission Microscopy. *Plasmonics* **13**, 239–246 (2018)
8. Nelayah, J. *et al.* Mapping surface plasmons on a single metallic nanoparticle. *Nat Phys* **3**, 348–353 (2007)
9. Schmidt, F.-P. *et al.* Dark plasmonic breathing modes in silver nanodisks. *Nano Lett* **12**, 5780–3 (2012)
10. Chu, M.-W. *et al.* Probing Bright and Dark Surface-Plasmon Modes in Individual and Coupled Noble Metal Nanoparticles Using an Electron Beam. *Nano Lett* **9**, 399–404 (2009)
11. Dorfmueller, J. *et al.* Fabry-Pérot Resonances in One-Dimensional Plasmonic Nanostructures. *Nano Lett* **9**, 2372–2377 (2009)

3. As the authors are introducing a new NSOM measurement protocol, the performance of the method is very important. They do a very nice job letting us know how long a complete measurement (with 100 angles) takes, and showing how taking different number of measurements affects the final reconstruction (Fig. 3). What I am missing, in this vein, is a similar discussion on the measurement resolution. In this work, data is taken with 25 nm steps (line 139), which is well below the resolution of the image formation process (see Light, Science and Applications, 8, 28 (2019)). I think a study of the reconstructed image as a function of the measurement step size is hence very important; it seems likely to me that they are currently over-sampling and therefor needlessly increasing their measurement time. To me, a plot of the reconstruction fidelity of each mode as a function of step-size is missing.

This is a good suggestion. Similar to the analysis for the number of illumination angles shown in Fig. 3, we conducted mode mapping analysis depending on the scanning step size (Fig. R8). We

constructed FNTMs with the scanning step of 50 nm, 75 nm and 100 nm from the original FNTM taken at 25 nm scanning step and obtained their respective eigenmodes. Up to the 50 nm scanning step, the nodes of the higher order modes were well visible. However, the image pixelation became so pronounced for the further increase in the scanning step that the fine mode structures were lost, especially at higher order modes. Considering the trade-off relation between the acquisition time and scanning step, the scanning step of 50 nm can also be a good choice because the matrix acquisition time can be reduced by almost a factor of four.

Figure R7. Near-field eigenmodes of double nano-slits depending on the scanning step of the NSOM probe. Eigenmodes for the scanning steps of 100 nm, 75 nm, 50 nm, and 25 nm are shown. The white rectangles outline the boundaries of the nano-slits. Scale bar, 150 nm.

We added the following sentence to the revised manuscript, and the analysis shown in Fig. R8 was added to Supplementary Section I.4.

“It is noteworthy that the scanning step of the NSOM probe (25 nm) was much finer than the resolving power set by the probe aperture diameter (150 nm)¹ to ensure high mode reconstruction fidelity and minimize the pixelation artifact. From the separate analysis, we confirmed that the scanning step of 50 nm was good enough to map all the observed modes.”

1. Feber, B. le, Sipe, J. E., Wulf, M., Kuipers, L. & Rotenberg, N. A full vectorial mapping of nanophotonic light fields. *Light Sci Appl* **8**, 28 (2019)

4. Around lines 51-53, the authors argue that conventional NSOM would only map the dominant mode at the excitation wavelength. This argument only holds for nanoparticles. For propagating modes a Fourier Transform can be used to separate the different modes/harmonics.

In case when multiple modes are superposed at a single excitation wavelength, it is not possible, in general, to separate individual modes from the recording of a single complex-field NSOM imaging. One can selectively map the individual modes when they are spectrally distinct¹⁻³. Or, if the modes have spatially distinct distribution, one can identify individual modes by the spectral response of local near field⁴. We are not sure how the Fourier transform of the conventional NSOM image can lead to the separation of different modes. To make our claim legitimate in a more general sense, we revised the sentence as follows.

“This is an important addition of information to the conventional NSOM, in which only the integrated signal from all the contributing modes at the excitation wavelength is detected.”

1. Alonso-Gonzalez, P. et al. Real-Space Mapping of Fano Interference in Plasmonic Metamolecules. *Nano Letters* **11**, 3922-3926 (2011)

2. Alfaro-Mozaz, F. J. et al. Nanoimaging of resonating hyperbolic polaritons in linear boron nitride antennas. *Nature Communications* **8** (2017)
3. Tamagnone, M. et al. Ultra-confined mid-infrared resonant phonon polaritons in van der Waals nanostructures. *Science Advances* **4** (2018)
4. Nelayah, J. et al. Mapping surface plasmons on a single metallic nanoparticle. *Nat Phys* **3**, 348–353 (2007)

5. On line 72, the authors state that the properties of nano-slits are well understood. This statement begs for references.

We added the following references elucidating the formation of hybridized eigenmodes by the mode coupling between the nano-slits.

1. Park, Y., Kim, J., Roh, Y.-G. & Park, Q.-H. Optical slot antennas and their applications to photonic devices. *Nanophotonics-berlin* **7**, 1617–1636 (2018)
2. Lee, S., Park, Y., Kim, J., Roh, Y.-G. & Park, Q.-H. Selective bright and dark mode excitation in coupled nanoantennas. *Opt Express* **26**, 21537–21545 (2018)

6. On line 117, the authors state that the phase of the reference field is taken to be constant. But the authors measure across different resonances, each which will contribute a different phase spectrum. Or, do they mean constant in space, but not in frequency? In either case, this should be clarified.

What we meant is that the phase of the reference field is constant in space. We revised the sentence to make this point clear. We also added the near-field phase map of the reference wave to Supplementary Section III.2.

“The phase of $E_R(x, y)$ was assumed to be flat in space, which is typical for the symmetrically driven subwavelength nanostructures (see Supplementary Section III)”

7. On line 140, the authors state that they hold the tip about 1 nm away from the sample. This is much closer than the 10-25 nm typically reported. How do the authors know that they are at this distance and how do they manage to scan at this height?

There was a mistake in translating manufacturer’s specification. In fact, the distance between the tip and sample was maintained to be 20 nm. We corrected this error in the revised manuscript.

8. I read the statement on line 152-3 that the image depends on k^{in} because it depends on k^{in} , which is meaningless. Please rewrite so that this means something.

We revised the sentence to make its meaning specific.

“With the increase of $|\vec{k}^{in}|$, the phase difference of the excitation wave at the two nano-slits is increased such that the way the eigenmodes are superposed varies with \vec{k}^{in} .”

REVIEWERS' COMMENTS:

Reviewer #1 (Remarks to the Author):

The revised version of the manuscript entitled "Near-field transmission matrix microscopy for mapping high-order eigenmodes of subwavelength nanostructures" resolves most of the issues pointed out by the reviewers. Overall, the method is well explained, well supported by additional measurements and calculations. The originality of the method is clearly supported. I therefore tend to recommend the paper for publication.

However, the motivation presented by the authors in the introductory paragraphs, claiming that the method is important for optimizing the performance of nanophotonic devices, may be an overstatement and seems to lack credibility. In this respect, the motivation is lacking a concrete example supporting the unique capability of the method developed by the authors to improve the nanostructure design. Electromagnetic response of nanostructures can be understood by combining (already existing) experimental inputs (e.g. particle morphology, dielectric function) with well-established theoretical methods. Furthermore, the methodology to design optical nanostructures will strongly depend on the purpose of the particular device and generalizations are therefore not appropriate.

In my view, the method is a useful nontrivial extension of existing near-field microscopic techniques which have mostly applications in the area of basic research. I would therefore strongly recommend that the authors reconsider the focus of the introduction towards more realistic applications of the method.

Reviewer #2 (Remarks to the Author):

All my comments/questions have been answered.

I have no further comments, although I still think this research is too specific for Nature communications.

Reviewer #3 (Remarks to the Author):

The authors have nicely addressed my concerns, and those of my fellow referees, and I find the revised manuscript much clearer. I would recommend publication as is.

Just for the authors information, in regards to my previous comment 4. A waveguide may support multiple modes, at the same frequency, and in general they will have different k-vectors (that is effective wavelengths). The Fourier transform of a complex field map of such a waveguide will reveal the separate modes that, after filtering, can be reconstructed separately.

Reviewer #1

The revised version of the manuscript entitled "Near-field transmission matrix microscopy for mapping high-order eigenmodes of subwavelength nanostructures" resolves most of the issues pointed out by the reviewers. Overall, the method is well explained, well supported by additional measurements and calculations. The originality of the method is clearly supported. I therefore tend to recommend the paper for publication.

However, the motivation presented by the authors in the introductory paragraphs, claiming that the method is important for optimizing the performance of nanophotonic devices, may be an overstatement and seems to lack credibility. In this respect, the motivation is lacking a concrete example supporting the unique capability of the method developed by the authors to improve the nanostructure design. Electromagnetic response of nanostructures can be understood by combining (already existing) experimental inputs (e.g. particle morphology, dielectric function) with well-established theoretical methods. Furthermore, the methodology to design optical nanostructures will strongly depend on the purpose of the particular device and generalizations are therefore not appropriate.

In my view, the method is a useful nontrivial extension of existing near-field microscopic techniques which have mostly applications in the area of basic research. I would therefore strongly recommend that the authors reconsider the focus of the introduction towards more realistic applications of the method.

We agree with the reviewer in that the applicability of the proposed method is generalized a bit more than necessary. We therefore eliminated the sentence, "Furthermore, the fabrication is more likely to be off from the design," in the introduction. And we revised the related paragraph as follows to tone down the device design aspect and to more emphasize our method's benefits in the fundamental research.

"With advancements in fabrication technology, there have been growing interest in designing smaller nanostructures and integrating them at a greater density for merging multiple functionalities within a smaller chip. This often makes it necessary to engineer the hybridized modes having distinct resonances with minimal cross-talks by properly engineering the coupling among the modes of the constituent nanostructures. However, mode hybridization is more complicated because the modes of the basic building blocks such as nanoparticles, nanorods and nano-slits are spectrally broadened and spatially overlapped with the reduction of their size. In this respect, most of the existing NSOM modalities are not well suited for elucidating the detailed mode formation mechanism. They tend to map either the dominant modes or integration of all the modes at the given excitation/detection wavelength. To better understand the formation of the hybridized modes in the small-scale nanophotonic devices, it will be important to have experimental tools that can separately map the multiple superposed modes."

Reviewer #2

All my comments/questions have been answered.

I have no further comments, although I still think this research is too specific for Nature communications.

We understand the reviewer's concern for the applicability of the proposed method. In fact, it usually takes time to make the most of the emerging methodologies. As our method is clearly adding new capability to the existing NSOM, we believe that the near-field imaging community will capture its

importance and broaden the range of applications in the coming years.

Reviewer #3

The authors have nicely addressed my concerns, and those of my fellow referees, and I find the revised manuscript much clearer. I would recommend publication as is.

Just for the authors information, in regards to my previous comment 4. A waveguide may support multiple modes, at the same frequency, and in general they will have different k -vectors (that is effective wavelengths). The Fourier transform of a complex field map of such a waveguide will reveal the separate modes that, after filtering, can be reconstructed separately.

We deeply appreciate the reviewer's valuable suggestions and support of our study. We now understand the comment that the reviewer remarked on the extraction of individual modes by the spatial Fourier transform. We agree with the reviewer that this method can be applied to many systems where the modes have distinct wavevectors. From this point of view, our method works even when the wavevectors of the modes are overlapped.